# New Therapeutic Approaches for Allergy: A Review of Cell Therapy and Bio- or Nano-Material-Based Strategies

**DOI:** 10.3390/pharmaceutics13122149

**Published:** 2021-12-14

**Authors:** Juan L. Paris, Paz de la Torre, Ana I. Flores

**Affiliations:** 1Allergy Research Group, Instituto de Investigación Biomédica de Málaga-IBIMA, 29010 Málaga, Spain; juan.paris@ibima.eu; 2Andalusian Centre for Nanomedicine and Biotechnology-BIONAND, 29590 Málaga, Spain; 3Grupo de Medicina Regenerativa, Instituto de Investigación Sanitaria Hospital 12 de Octubre (imas12), 28041 Madrid, Spain; torre-merino.paz@h12o.es

**Keywords:** allergy, cell therapy, biomaterials, nano-materials, immunotherapy

## Abstract

Allergy constitutes a major health issue due to its large prevalence. The established therapeutic approaches (allergen avoidance, antihistamines, and corticosteroids) do not address the underlying causes of the pathology, highlighting the need for other long-term treatment options. Antigen-specific immunotherapy enables the long-term control of allergic diseases by promoting immunological tolerance to the allergen. However, efficacious immunotherapies are not available for all possible allergens, and the risk of undesired reactions during therapy remains a concern, especially in patients with severe allergic reactions. In this context, two types of therapeutic strategies appear especially promising for the future in the context of allergy: cell therapy and bio- or nano-material-based therapy. In this review, the main strategies developed this far in these two types of strategies are discussed, with several examples illustrating the different approaches.

## 1. Introduction

An allergic reaction is the useless response of the immune system toward a harmless substance (allergen) and is the consequence of a failure in the development of tolerance. It involves components of the innate and the adaptive immune responses as type 2 innate lymphoid cells (ILC2s), mast cells (MCs), eosinophils, and basophils, as well as CD4+ T helper type 2 (Th2) cells and IgE producing B cells [1]. Allergy develops as a sequential process that begins with the sensitization to the allergen and follows by an effector phase in which the clinical symptoms appear.

The sensitization starts from allergen capture by dendritic cells (DCs) resident in the airway, skin, or gut. DCs internalize and process the antigen into peptides and present them to naïve CD4+ T cells, which become allergen-specific Th2 cells. Th2 cells, through surface molecules and secretion of type 2 cytokines (IL-4, IL-5, and IL-13), promote Ig isotype switch in B cells, which differentiate to plasma cells, producing allergen-specific IgE antibodies. Allergen-specific memory Th2 and B cell populations expand and are stimulated upon re-exposure to the allergen [2].

The effector phase starts within seconds or minutes of the re-exposure and is the consequence of the cross-linking of IgE to receptors in basophils and MC, which triggers cell activation and the release of a wide array of mediators, such as histamine, heparin, proteases, leukotrienes, and cytokines that cause inflammation and allergic symptoms [3].

The high prevalence and associated burden of allergic disorders make them a global health problem. The European Academy of Allergy and Clinical Immunology (EAACI) estimated that 150 million Europeans suffered from an allergy in 2016. The steady increase of cases suggests that in the foreseeable future, more than half of the population, especially in western countries, will suffer from an allergy [4]. From a therapeutic point of view, avoiding the allergen would be the first and main recommendation, but once the allergic reaction has occurred, medications such as corticosteroids, antihistamines, or α2 adrenergic agonists are used. These symptomatic medications are intended to reduce inflammation, but their long-term use is not highly recommended. In recent years, by the knowledge of the cellular and molecular bases of allergy, new therapeutic approaches are emerging, which attempt to act on the underlying pathogenic mechanism of allergy and provide long-term effects. Promoting tolerance to allergens is the goal of these approaches. Some of these treatments are already being used to treat the allergic patient, while others are new strategies that are being studied at the preclinical level. On the one hand, once the main allergy players are known, some treatments attempt to modulate the immune reaction by redirecting the cellular response to a tolerogenic state or by exogenously transferring tolerogenic cells or their secretome. On the other hand, allergen-specific immunotherapy (AIT) aims to achieve allergen tolerance by exposing the patient to small, increasing doses of an allergen (Figure 1).

In the context of AIT, strategies based on bio- or nano-materials have attracted great attention in recent years. In particular, the use of nano-materials for biomedical applications (nanomedicine) has attracted great interest in recent years [6]. The early development of the nanomedicine field focused mainly on the use of drug delivery nanocarriers for anticancer therapy [7]. This focus on cancer nanomedicine was mainly driven by the accumulation of nanoparticles in tumor tissue due to what was called the enhanced permeability and retention (EPR) effect [8,9]. The EPR effect was then used as the main rationale behind cancer nanomedicine and was also commonly referred to as the “passive targeting” principle. Further progress in the field led to the development of “active” targeting approaches, where the surface of nanoparticles was decorated with different moieties driving enhanced uptake by certain target cell types [10]. Although the physiological mechanism behind the passive accumulation of nanoparticles in tumors has been called into question [11], the tools that have been devised by the cancer nanomedicine community are now being exploited for other purposes. Among the most promising applications currently being evaluated is the use of nanoparticles for immunotherapy in cancer [12] as well as in other diseases such as autoimmunity and organ transplantation [13,14,15]. Here, we will focus on a promising (although still relatively underexplored) use of bio- and nano-material for immunomodulation: allergy therapy. These bio- and nano-material-based strategies include approaches in which the material interacts directly with immune cells to provide a therapeutic effect, as well as schemes in which the material acts as a delivery system for an allergen, a drug, or a combination of both.

This review does not aim to provide a comprehensive list of all the novel strategies for allergy therapy that can be found in the literature. The main purpose of this work is to describe the main types of cell-, bio-, or nano-material-based strategies designed to acquire tolerance as allergy treatment (Figure 2). Relevant examples will be highlighted to illustrate each of the main strategies within each of these areas.

## 2. Cell-Based Therapeutic Actions toward Tolerance State in Allergy

### 2.1. Key Players Involved in Allergy and Their Use for Cell Therapy

#### 2.1.1. Dendritic Cells (DCs)

DCs are the most specialized antigen-presenting cells (APC) and are considered the connection point between innate and adaptive immunity. DCs develop from bone marrow progenitor cells and complete their differentiation in the periphery, being present in the skin, mucous membranes, and tissue parenchyma. They act as sentinel cells that initiate and regulate the immune response. DCs take charge of trapping antigens that penetrate from the external environment and deliver them to the lymphatic organs, where they induce the activation, differentiation, and expansion of naive CD4+ and CD8+ T cells. Depending on their maturation/activation status and the lineage markers expression, DCs have also been shown to induce a tolerance state by inhibition of T-cell responses through apoptosis, anergy, generation of regulatory T (Treg) cells, and secretion of immunosuppressive cytokines [16]. In allergic reactions, the use of naturally-occurring or induced tolerogenic dendritic cells (tDCs) may be a prophylactic and therapeutic option to control exacerbated immune responses [17]. The techniques of isolation and reprogramming of DCs toward tolerogenic profiles are well characterized, the latter including treatment with some cytokines such as transforming growth factor (TGF)-β, interleukin (IL)-10, and/or pharmacological agents such as corticosteroids or rapamycin [18]. There are studies showing that bone marrow-derived DCs exposed in vitro to IL-10 and transferred to ovalbumin (OVA)-induced allergic mice reduced airway hyperresponsiveness, eosinophilia, and Th2 cytokine release [19]. Equally, bone marrow-derived DCs genetically engineered to express IL-10 suppressed experimental asthma induction [20]. Although favorable effects have been reached in clinical trials, tDC adoptive cell therapy remains challenging. Isolation, purification, and culture of DCs is an expensive protocol, and the resultant tDCs are a bespoke patient product due to histocompatibility concerns. As an alternative to this ex vivo generation of tDCs, the in vivo tolerogenic payload delivery to DCs is proposed, and nanoparticles offer the possibility of simultaneously supplying antigens, together with tolerogenic agents, to be delivered to DCs directed by DC-specific target ligands for generating antigen-specific tolerogenic DCs [21]. This strategy will be further explored in a later section in this article.

#### 2.1.2. B lymphocytes and Immunoglobulin E

B lymphocytes play a pivotal role in allergy by their terminal differentiation into allergen-specific immunoglobulin (Ig)E-secreting plasma cells. IgE is the antibody responsible for allergic reactions and owes its name to the presence of epsilon chains in its structure. IgE is the least present immunoglobulin in blood but is the principal antibody against parasite diseases, mainly those caused by helminths. In countries where the improvement of hygienic conditions has made parasite diseases into very rare events, the immune system of some people synthesizes high levels of IgE against allergens that are harmless for most people. After allergen contact, responding T cells, by means of CD40L and cytokines, can engage CD40 on B cells and switch them to produce IgE antibodies. Subsequently, it is described that B cells acting as antigen-presenting cells (APCs) through the low-affinity receptor for IgE, CD23, may control allergen-specific T-cell activation through IgE-facilitated allergen presentation [22]. It has been experimentally demonstrated that chronic exposure to allergens may drive the accumulation of long-lived IgE plasma cells in the bone marrow, which maintains allergic memory over time [23]. On the contrary, there is a subset of B cells that perform an inhibitory role over immune response through surface molecules and the secretion of cytokines. The role of these regulatory B (Breg) cells in allergy suppression has been largely evidenced. Breg immunosuppressive capacity is often mediated through IL-10 secretion, although IL-35 and TGF-β have also been involved by suppressing effector T-cell responses and inducing Treg cells [24]. In addition, forkhead winged helix transcription factor forkhead box p3 (Foxp3)-expressing B cells have been identified and may act as regulatory B cells [25]. The proportion of Breg cell subsets within B cells was found to be lower in patients with allergic rhinitis and allergic asthma, and allergen immunotherapy enhanced the frequency of IL-10-producing antigen-specific B cells [5]. Adoptive transfer of Breg cells was shown to normalize airway inflammation and lung function in a murine model of allergic asthma in an IL-10-dependent manner [26].

#### 2.1.3. Mastocytes

After susceptible subjects are sensitized to an allergen, circulating allergen-specific IgEs bind to mast cells (MCs) via high-affinity IgE receptors (FcεRI) on the cell surface. Mast cells reside within the tissues interfacing with the external environment as skin and mucosal membranes, and also in vascularized tissues next to nerves, vessels, and glands. IgE binding triggers MC degranulation with the release of biologically active preformed mediators involved in the inflammatory local or systemic response to allergens [27].

Bringing down serum levels of IgE is a mechanism of reducing allergic manifestations. Omalizumab, first approved by the USA in 2003, is a humanized monoclonal antibody that binds serum IgE and blocks both early and late-phase reactions to allergen challenge, with demonstrated effectiveness and safety, reducing symptoms, frequency of exacerbations, and steroid requirement [5]. In addition, it is described that exosomes isolated from the MCs secretome can suppress allergic reactions by binding to IgE [28]. Exosomes are extracellular nano-sized structures, cell membrane–delimited, carrying a range of biomolecules associated with the cell type of origin [29]. It has been described that MCs exosomes harboring high-affinity IgE receptors (FcεRI) bind free serum IgE and decrease IgE levels in a mouse model of allergic asthma. This results in the inhibition of MC activation and the modulation of airway inflammation so that MCs exosomes have been proposed as a novel anti-IgE agent.

#### 2.1.4. Regulatory T Cells (Tregs)

The development of immune tolerance strongly relies on regulatory T cells (Treg), which have a main role in the prevention of autoimmune diseases, graft rejection, and allergic reactions [30]. Tregs are a subpopulation of CD4+ T cells expressing CD25, the alpha chain of the IL-2 receptor, and two main subgroups are described: natural Tregs, arisen in the thymus (named nTreg, or also tTreg), which mediate tolerance to self-antigens; and adaptive or induced Tregs (iTreg, also named peripheral or pTreg), which are particularly enriched in the gastrointestinal tract and lungs and promote tolerance to foreign antigens [31]. These cells can suppress immune responses by several means, such as inhibiting T cell proliferation and cytokine production, regulating B cell responses, and modulating innate immunity. There are no exclusive markers for Tregs, but both types express the Foxp3, which is essential to their function [32].

The role of Tregs in sustaining immune tolerance to allergens is clearly evidenced, and it is proposed that allergic reactions are the result of the dysfunction of allergen-specific iTregs in genetically allergy-predisposed people [33]. Foxp3 deficiency has been described to lead to a severe immune dysregulation syndrome characterized by autoimmunity, severe allergic manifestations, and fatal outcomes in mice and human beings [34]. In addition, it has been described that a deregulated function of Tregs, causing a Th2 immune response predominance, is involved in the development of asthma [35]. TGF-β was identified as the main driver of iTreg differentiation and its depletion causing exacerbated symptoms in an experimental model of asthma [36]. It is evidenced that there are several mechanisms by which Treg cells control allergic reactions. By direct cell-to-cell contact with mast cells, Tregs inhibit MC degranulation preventing the release of mediators such as histamine and blocking the hypersensitivity response [37]. Tregs suppress B cells Ig production in a cell-cell contact manner [38]. Furthermore, there is evidence that interleukin (IL)-10, a main player in the iTreg defense against allergic reactions, results in B cells switching to IgG4 production to the detriment of IgE promoting the tolerogenic environment [39]. Tregs also act on DCs, by the downregulation of their surface molecules CD80/CD86 and thus blocking a subsequent allergen-specific Th2 cell response [40].

In the treatment of cases of severe food allergy by induced oral tolerance, clinical response correlates with the restoration of Treg number and function. Children who had outgrown a milk allergy by oral tolerance induction showed a higher frequency of milk-protein-specific Tregs [41]. Oral immunotherapy leading to peanut tolerance in allergic patients correlated with an increased number of circulating allergen-specific iTreg cells [42]. On the contrary, in peanut-sensitized mice, the depletion of Tregs by means of anti-CD25 monoclonal antibody resulted in impaired oral tolerance and strengthened allergic reaction [43].

Therapeutic approaches in allergic diseases have been designed to benefit from Tregs. It includes the adoptive transfer of allergen-specific T regulatory cells to improve tolerance in experimental models of food allergy and asthma. In the case of ovalbumin allergy, the transfer of Tregs reduced airway hyper-reactivity and eosinophil recruitment [44] and suppressed anaphylactic responses [45]. Some mediators, such as IL-10 and IRF-4, have been described to support the role of Treg cells in suppressing Th2-driven mucosal inflammation [46]. Some challenges remain in the development of therapy by Treg cells, given that adoptive transfer of exogenous Tregs has sometimes implied serious risks and even fatal consequences [47].

Besides this, adoptive cell therapies using naturally occurring allergen-specific Tregs, chimeric antigen receptor (CAR)-T cells, have been used. CAR-T cells are manipulated lymphocytes that express genetically engineered T cell receptors (TCR) with or without co-stimulatory molecules. For the treatment of allergy, CAR-T cells can be genetically engineered to contain a regulatory domain (i.e., CAR-Treg). Directing Tregs by a CAR toward the target tissues offers a more precise treatment. In mice, experimentally induced allergic asthma antigen-specific CAR-Tregs were designed to recognize the intended antigen and to redirect these cells toward airways. CAR-Treg cells can accumulate and become activated in the inflamed airways of the asthmatic lung, where they can control allergic inflammation [48]. CAR-T cell products are a clinical reality already used in cancer immunotherapy to treat hematologic malignancies. Nevertheless, this therapy can sometimes cause serious or even life-threatening side effects as a consequence of the excessive release of cytokines (“cytokines storm”), neurotoxicity, cytopenia, and infection complications, among others [49]. Real efforts are made to understand the causes of toxicity and to design strategies to avoid it without affecting therapy efficacy.

#### 2.1.5. Exosomes

Exosomes are essential players in cell-to-cell communication since they contain bioactive molecules such as proteins, lipid mediators, and nucleic acids, particularly microRNAs (miRNA), which once released into the extracellular space modulate the activity of acceptor cells. miRNAs are short RNA sequences (around 19–22 nucleotides) that regulate the expression of genes encoding target proteins. They generally bind to the 3′-UTR (untranslated region) of their target mRNAs and repress protein synthesis by destabilization of the mRNA and/or translational silencing [29]. In allergy, a relevant role of exosomes and miRNAs in the development and maintenance of the pathological state is anticipated (Figure 3). In asthmatic mouse models, exosome secretion into bronchoalveolar lavage fluid (BALF) was elevated upon allergen exposure [50]. It is known that several cell types release exosomes whose specific content may play a central role in the orchestration of the allergic reaction [51]. Exosomes from DCs carry aeroallergens and contribute to allergic inflammation, as well as MHC molecules being able to stimulate T-cell responses. Exosomes from MCs stimulate purified B cells to produce IgE through a T-cell independent mechanism. B-cell-derived exosomes carrying MHC class II, co-stimulatory molecules (CD40, CD80, and CD86), and integrins can induce T-cell responses. Exosomes from activated T cells have been reported to deliver signals to mast cells, thereby enhancing mast cell activation. Furthermore, several miRNAs differentially expressed in exosomes isolated from asthmatic patients and healthy control subjects have been linked to airway inflammation [52].

Exosomes have also been described as players in tolerance acquisition. It is proposed that oral tolerance to food is mediated by exosomes with immuno-regulatory functions (tolerosomes), which express major histocompatibility complex class II molecules (MHCII) [53]. The antigen-specific tolerance begins in the gastrointestinal tract with the active sampling by the small intestinal epithelial cells (IECs) at the mucosal surface, the assembly with MHC II, and the loading onto tolerosomes, which are released into circulation from where they could be endocytosed by liver sinusoidal DCs [54]. In a murine allergy model, exosomes collected from the serum of mice that had been fed the allergen were transferred to recipient mice, which resulted in allergy protection [55]. This adoptive transfer of tolerosomes is one of the promising approaches in the use of exosomes for the control of allergic diseases. Exosomes are increasingly being recognized as a therapeutic tool because of their versatility, biocompatibility, and capacity to overcome biological barriers, and their potential use as drug delivery vehicles. In addition, exosome membranes have been used to provide biomimetic features to nanoparticles. Pei et al. presented a delivery platform consisting of exosome membrane-modified poly(lactic-co-glycolic acid) (PLGA) nanoparticles to carry a smart silencer into macrophage M2s in a mouse model of allergic asthma (AA) [56]. A smart silencer is a hybrid approach composed of small interfering RNAs (siRNAs) and antisense oligonucleotides (ASOs), which interfere with the expression of a given target gene. In this design, the smart silencer for DNA methyltransferase 3A opposite strand (Dnmt3aos), also known as long non-coding (lnc)-RNA, which regulates DNA methyltransferase 3A (Dnmt3a) expression, is used to prevent polarization of macrophage M2s, which play an important role in the onset of AA. It is shown that these biomimetic NPs effectively accumulate in the lungs and promote gene silencing that is accompanied by a reduction in the inflammatory cell infiltration degree into the airway.

### 2.2. Other Cell-Based Therapeutic Actions in Allergy

#### 2.2.1. Mesenchymal Stromal Cells (MSC)

Although not involved in the allergic reaction, mesenchymal stromal cells (MSC) have increasingly emerged as a potential treatment for allergic diseases. MSC are cells with tissue repair potential through their self-renewal and differentiation capacities. MSC are derived from the embryonic mesoderm layer and can be easily isolated and expanded in culture as fibroblast-like plastic-adherent cells from almost all adult and neonatal tissues [57]. Along with their regenerative capacity, MSC have shown immunomodulatory activities by exerting a suppressive effect on T cells, B cells, DCs, and natural killer (NK) cells, among others. MSC are a promising cell source for the treatment of autoimmune, degenerative, and inflammatory diseases. Indeed, MSC have been administered in autoimmune disorders, such as systemic lupus erythematosus, multiple sclerosis, or rheumatoid arthritis, and to control inflammation in GVHD, ulcerative colitis, and Crohn disease, among others [58]. In allergic disease models, MSC treatments control the exacerbated immune response and attenuate the clinical symptoms by means of several mechanisms [59]. As an example, a shift from Th2 to Th1 responses has been described in a mouse model of allergic airway inflammation after systemic injection of bone marrow-derived MSC [60]; likewise, MSC from adipose tissue contributed to the downregulation of Th2 responses and reduced eosinophilic inflammation in the nasal mucosa in a model of allergic rhinitis [61]. Unlike other interventions based on cells that are allergen-specific, MSC therapy modulates the immune system in an allergen-independent manner to potentially minimize the allergic reaction rather than desensitizing to a specific allergen.

As it is widely demonstrated, MSC have the ability to migrate toward inflamed sites after systemic administration. The expression of a variety of chemokine and cytokine receptors (CXCR1, CXCR2, CCR1, CCR2, etc.) in response to inflammation probably direct MSC to the damaged tissue. In this context, some methods to improve the trafficking and engraftment of MSC have been developed by means of genetically-induced overexpression of some signaling receptors [62]. In a mouse model of contact hypersensitivity (CHS), the overexpression of CXCR5 in MSC (MSC^CXCR5^) significantly increased the migrating ability of intravenously infused MSC toward CXCL13 expressed in the injured tissue. MSC^CXCR5^ strongly ameliorated CHS in the mice, as evidenced by decreased inflammatory cell infiltration in the irritated skin and pro-inflammatory cytokine production. MSC^CXCR5^ inhibited T cell proliferation and promoted T cell apoptosis [63].

Accumulating evidence informs the main therapeutic effects of MSC are exerted in a paracrine/autocrine form through secreted bioactive molecules. MSC are known to release various soluble factors that are responsible for the immunosuppressive response, including prostaglandin E2, indoleamine 2,3-dioxygenase (IDO), heme oxygenase-1 (HO-1); TGF-β, IL-10, leukemia inhibitory factor (LIF), human leukocyte antigen (HLA)-G5, nitric oxide (NO), TNF-stimulated gene 6 (TSG-6), interleukin (IL)-6, and interleukin 1 receptor antagonist [64]. The MSC secretome is used for therapeutic purposes as a cell-free medication in regenerative medicine, bypassing many limitations of MSC-based therapy, as undesired differentiation and potential activation of the allogeneic immune response. Exposing the MSC to different environmental conditions allows conditioning the secretome. It was revealed that the secretome from MSC exposed to inflammatory cytokines (TNF-α and IFN-γ) presents enhanced immunomodulatory properties. In allergic conjunctivitis, the treatment with conditioned media from TNF-α-treated MSC exerted an anti-allergic effect through the reduction of inflammatory cell infiltration in the conjunctiva and inhibition of B cells and MCs in a COX-2-dependent mechanism [65].

It is also possible to use extracellular vesicles as cell-free therapeutic agents with low immunogenicity and high biosafety. Small extracellular vesicles (sEV) derived from MSC have been used in a mouse model of asthma. The systemic administration of MSC-sEV produced a reduction of ILC2 levels, of inflammatory cell infiltration, as well as a reduction of mucus production in the lung, of Th2 cytokines, and an alleviation of airway hyperresponsiveness [66].

Exosomes secreted by bone marrow MSC were also found to exert an immunomodulation effect on peripheral blood mononuclear cells (PBMCs) of an asthmatic patient by means of IL-10 and TGF-β upregulation and thus promoting the proliferation and the immunosuppressive capacity of Tregs [67].

#### 2.2.2. Microbiome

The microbiome that microorganisms live on or within another organism has a relevant role in human physiology, contributing to the enhancement or impairment of metabolic and immune functions. It is considered a player in achieving tolerance. In the human body, it has been estimated that the number of colonizing microbes is of the same order as the number of human cells [68]. These microorganisms inhabit and adapt to different niches in the body and, therefore, facultative anaerobes are dominant in the gastrointestinal and genitourinary tracts, whereas strict aerobes occupy the respiratory tract, nasal cavity, and skin surface. The relation of the microbiome to immune system maturation has been established by the discovery of major immunological abnormalities in germ-free mice [69], and among the proposed mechanism is, for example, the production of short-chain fatty acids by microbes growing on mucosa that promote the expansion and differentiation of Tregs [70]. Changes in levels and diversities of the microbiome have been found in patients with allergic diseases. The “hygiene hypothesis” formulated by David Strachan [71] proposes the protective role of early life exposure to microbes in allergy development. Dysbiosis in the skin, respiratory, or gastrointestinal tract, caused by diet or other environmental conditions, has been related to the activation of inflammatory pathways, and as an example, gut dysbiosis is increasingly being associated with asthma [72]. The administration of probiotics, prebiotics, and/or a combination of both pursues restoring altered microbiome functionality and is thought of as an adjuvant treatment in specific immunotherapy and, in the case of allergy, can reduce the morbidity and duration of allergy symptoms [73]. In the oral administration of live bacterial cells for therapy, effective delivery to the intestine could be enhanced by the use of different encapsulating biomaterials that protect them from the acidic environment of the stomach (reviewed in Liu et al., 2021) [74]. These formulations based on alginate, chitosan, or pectin, among others, are validated in vitro for resistance to a simulated gastric solution without loss of viability of contained microbes.

## 3. Bio- and Nano-Material-Based Strategies for Allergy Therapy

AIT is the only allergy treatment that can provide long-term effects since it tries to address the underlying pathological mechanism, changing the immune response to the allergen from a Th2 response towards a Th1 or a regulatory (Treg) response [75]. However, in some cases, especially in severe allergy reactions, the administration of the allergen during AIT (even if it is performed at extremely low doses) can trigger an undesired immune response that can be potentially life-threatening. For this reason, alternative allergen administration strategies have been developed to maintain the therapeutic potential of AIT while decreasing the associated risks. One of the most common strategies is the use of hypoallergenic immunotherapy agents, which have been modified to keep the immunotherapeutic potential from the original allergen, while drastically reducing the likelihood of producing an undesired allergic response [76]. An alternative strategy is to use different delivery vehicles that can concentrate the allergen in target cells, organs, or tissues, decreasing the dose of allergen needed for the therapy and, therefore, diminishing the risk of undesired reactions [77,78,79,80]. These delivery vehicles are usually bio- and nano-materials whose properties can be tuned for each specific application, depending on the cargo to be loaded inside, the route of administration, and the target cells or organs. Additionally, these materials can also inherently act on the allergic response (either positively or negatively) without the need for a therapeutic cargo, depending on their composition, size, and other physicochemical characteristics, so some could be used directly as a therapeutic option in allergy [81]. Therefore, there are three main strategies in which bio- and nano-materials can be used as a therapeutic tool in allergic diseases:Using bio- and nano-materials with a direct effect on cells involved in the allergic response.Use of bio- and nano-materials as allergen delivery vehicles for immunotherapy.Use of bio- and nano-materials as co-delivery systems containing the allergen and immunomodulatory molecules.

### 3.1. Using Bio- and Nano-Materials with a Direct Effect on Cells Involved in the Allergic Response

In this therapeutic strategy, the effect is caused directly by the interaction of the material with a certain cell type involved in the allergic response, without any cargo being delivered from the material. Therefore, the administration route and the therapeutic potential of each particular strategy will depend on the participating cell type. For example, materials that interact with cells involved in the sensitizing stage of allergic diseases will possess a principally prophylactic potential since their effect will probably be very limited once sensitization has already occurred. One of the processes that can be affected by this type of approach is allergen processing. Allergens display a particular processing kinetic, even if they share structural characteristics with non-allergenic proteins. The Allergenicity of an antigen is reflected by increased resistance for endolysosomal processing. High allergenic potential of Bet v 1 (major birch pollen allergen) has been attributed to its limited susceptibility for proteolytic degradation in DCs [82]. Nevertheless, decreasing antigen processing by means of the use of nanoparticles may switch DCs toward a tolerogenic state. DCs treated with polyvinyl alcohol-coated super-paramagnetic iron oxide nanoparticles (PVA-SPIONs) showed a decrease in antigen processing, expression of MHCII, and stimulation of CD4+ T cells in vitro, suggesting an intrinsic capacity of PVA-SPIONs for immune-modulation affecting DCs function [83]. However, it is worth noting that in the case that a similar prophylactic approach was successfully translated to the clinic, its use would be limited to high-risk individuals exposed to specific allergenic substances.

On the other hand, materials can interact directly with effector cells involved in triggering the allergic response, such as MC or basophils. As an example of this, we can find the work of Ryan et al. [84], where fullerenes were shown to exhibit a direct effect on MC behavior in vitro, decreasing the IgE-induced release of mediators. Furthermore, when this material was evaluated in a murine model of anaphylaxis, fullerene administration also partially prevented the decrease in body temperature and the release of histamine. Approaches similar to this have greater applicability in the clinic, as they could potentially be used for a wide range of allergies in which different allergenic proteins are involved. However, this material could be difficult to translate into the clinic due to some safety concerns of carbon-based materials that must be administered systemically (as is the case here) [85]. In any case, the limitations of this type of treatment are the same as for available treatments where only the effects of an individual reaction are addressed (such as with antihistaminic drugs), since there is no long-term effect of the treatment, and re-exposure to the allergen will re-induce a reaction. For this reason, most of the studies being carried out in bio- and nano-materials for allergy therapy are being focused on developing different AIT strategies involving these materials as delivery systems.

### 3.2. Use of Bio- and Nano-Materials as Allergen Delivery Vehicles for Immunotherapy

The use of a delivery vehicle can improve AIT efficacy and reduce the side effects by several mechanisms [86,87,88]. Firstly, by targeting nanocarriers towards specific cell types, the needed dose of allergen is decreased. Secondly, previously unsuitable administration routes or types of cargo can be employed for AIT with the help of these delivery vehicles. For example, a labile protein could be administered orally if included within a protective material, or a nucleic acid could be used for AIT if a carrier is specifically designed to enable its intracellular release in target cells. Thirdly, selective delivery to target organs or tissues can also be achieved, such as targeting tolerogenic organs. Through these works, it becomes clear that the route of administration of the immunotherapy agent is critical for its efficacy. Examples of each of these types of strategies are shown below to illustrate the broad range of possibilities available when combining AIT with bio- and nano-materials, as well as to reveal unexplored options for further research.

The first aspect to consider when formulating nanoparticles for AIT is the type of interaction between the allergen and the nanocarrier. Two main options are available: either the allergen is chemically conjugated to the nanocarrier (mainly on the particle surface), or the allergen is physically entrapped (encapsulated) within the nanocarrier. An important early work that investigated these options [89] reported the effect of three different ovalbumin (OVA)-carrying nanoparticles: non-biodegradable polystyrene nanoparticles with conjugated OVA, biodegradable PLGA nanoparticles with conjugated OVA, and PLGA nanoparticles with encapsulated OVA. The results obtained in vivo showed that OVA-conjugated polystyrene nanoparticles were effective as a tolerogenic prophylactic agent, but they induced anaphylaxis when administered to pre-sensitized animals. On the other hand, OVA-conjugated PLGA nanoparticles were also effective as a prophylactic treatment, and furthermore, they did not induce anaphylaxis in pre-sensitized mice and could partially inhibit Th2 responses (but not airway inflammation) when used as a therapeutic agent. This result highlights that the composition of the nanoparticles used for AIT and, more importantly, their biodegradability constitutes a major parameter driving their effect in vivo. Remarkably, OVA-encapsulated PLGA nanoparticles inhibited Th2 response and airway inflammation in mice, both prophylactically and after sensitization. These results indicate that the best option to develop nanoparticles for AIT is the use of biodegradable nanoparticles encapsulating the allergen.

Based on the results mentioned above, the simplest strategy to achieve material-driven AIT is to encapsulate allergenic proteins within nanoparticle formulations. Formulating the allergen within a particulate form can by itself ease its interaction with certain immune cells and increase the therapeutic immune response. Nanoparticles with many different compositions might be used for this application (lipidic [90], inorganic [91], polymeric [92]), although the most common material found in the literature for this application is polymeric nanoparticles. As an example of this line of work, Pohlit et al. developed acid-labile polymer-lipid nanoparticles loaded with allergens, including grass pollen allergen and house dust mite allergen [93]. When these nanoparticles were incubated with DCs in vitro, they did not induce maturation of the DCs, but they were capable of inducing the immune response of co-cultured T cells. Rebouças et al. prepared poly(anhydride) nanoparticles loaded with peanut proteins for in vivo allergy immunotherapy [94]. After intradermal particle administration, a mixed Th1/Th2 response was observed, with a more pronounced Th1 response when spray-dried nanoparticles were used compared to lyophilized ones. This response would be more suitable for allergen immunotherapy and was characterized by lower IgE and IL-5 levels and higher IFN-γ production. These results highlight that other formulation parameters may have a critical impact on therapeutic efficacy, in addition to the selection of cargo and particle composition.

Besides directly using the allergenic protein as the therapeutic cargo, a nucleic acid cargo can also be used, such as mRNA of plasmid DNA encoding the target allergenic protein. By using nucleic acid cargo, several advantages can be achieved. On the one hand, after delivery to the target cells, the nucleic acid can induce the expression of the therapeutic protein for some time, so the dose of the protein produced can easily surpass what could have been directly delivered to the same target cell by an analogous carrier particle. On the other hand, if there is an off-target release of the cargo outside of cells, the nucleic acid will not give rise to the production of the protein in those locations (out of the target region of the body). Since the specific IgE of the patient will only recognize the already-formed protein and not the nucleic acid that encodes it, this would greatly reduce the chances of triggering an undesired allergic reaction as a consequence of the treatment. As an example of this type of strategy, dendrosomes carrying plasmid DNA (containing the Betv 1a gene) were administered in the footpad of Balb/c mice in a prophylactic scheme (that is, prior to sensitization with the allergen rBetv1) [95]. The nanocarrier is critical in this type of strategy since it allows the intracellular delivery of the nucleic acid. The in vivo administration of these dendrosomes produced an increase in the IgG2a/IgG1 ratio and also an increase in IFNγ production in splenocytes, together with an inhibition of IgE and lower basophil degranulation. All these parameters indicate the induction of a stronger Th1 response.

The administration route is a critical parameter for successful AIT, both in terms of efficacy and safety. An advantage of the use of nanocarriers compared to administering the free therapeutic molecule is that it is possible to select the carrier composition and structure, as well as to tune its physicochemical properties to optimize the formulation for a specific route of administration. One common administration route used for allergy immunotherapy is the sublingual route. Among the main reasons to perform sublingual immunotherapy (SLIT) are the easy and non-invasive administration procedure, as well as the large number of immune cells in the oral mucosa, which can lead to a successful therapeutic effect. The use of DC-targeted nanoformulations allows for decreasing the amount of allergen necessary to obtain the desired effect while reducing undesired side-effects. In this context, aptamer-targeted PLGA nanoparticles loaded with OVA were developed to be captured by DCs in the sublingual mucosa during SLIT [96]. The aptamer-targeted OVA-loaded PLGA nanoparticle formulation used for SLIT was found to be the optimal treatment scheme when compared not only with free OVA administered sublingually but even with free OVA administered subcutaneously. The results showed a marked decrease in IgE, IL-4, and IL-17 levels, as well as a reduction in T cell proliferation and an increase in IFN-γ and TGF-β. This work highlights the great potential of DC-targeted nanoparticles to enable effective SLIT with greatly reduced allergen doses. As another example highlighting the potential of nanostructure-mediated SLIT for allergy, Rodriguez et al. reported the evaluation of mannose-modified dendrimers linked to a Pru p 3 peptide [97]. The mannose modification provided targeting towards DCs, and in a peach anaphylaxis mouse model, the authors found an optimal SLIT dose that could generate a strong long-term tolerance to the allergen [97]. Similar dendrimer-based structures for SLIT in a mouse model of Pru p 3-induced anaphylaxis were also previously reported by the same group using CpG-decorated dendrimers instead of mannose-modified ones [98].

Another interesting administration route is intranasal administration. In this context, Corthésy et al. designed a system for intranasal administration based on allergen-loaded gas-filled microbubbles [99]. When administered in a mouse model of allergic asthma, these microbubbles induced a tolerogenic response, with an increase in Foxp3+ CD4+ T cells (Tregs), IL-10, TGF-β, and the Th1 cytokine IFN-γ. Interestingly, this treatment did not just provide an effect when administered prophylactically, but also after the animals had already been sensitized to the allergen. In this context, the treatment could reduce the number of eosinophils and, decreasing the overproduction of mucus, improve lung functionality. Yet another interesting administration route for allergy immunotherapy is epicutaneous, since there is a large number of immune cells (such as Langerhans cells and other DCs) in the upper layers of the skin. In this regard, a particularly relevant biomaterial-based strategy is the use of microneedle (MN) patches. MN patches consist of arrays of needle-like structures of hundreds of µm in length that can be made of different materials (such as metals, polymers, or ceramics). Due to their short length, MN arrays do not produce any bleeding or pain when inserted into the skin since they do not reach the depth at which blood vessels and nociceptors are located. This eases the translation of MN-based therapeutics since they would be easy to adopt in populations where fear of needles can hinder successful adoption (such as in children). As an example of this strategy, Kim et al. prepared MN patches containing a *Dermatophagoides farinae* extract [100]. When this system was administered in vivo, the treatment produced a decrease in Th2 response and an increase in the Treg population (Foxp3+ CD4+ T cells) without significant side effects. The specific effects produced by the treatment included a decrease in IgE, epidermal thickness, and eosinophil count, together with an increase in IgG4, among other changes. When compared with subcutaneous immunotherapy (SCIT), the therapeutic effect observed with 10 µg of extract within MN patches was similar to that obtained with 100 µg of extract in SCIT, and clearly superior to that obtained with 10 µg of extract in SCIT.

Finally, besides adapting the materials for a particular route of administration, an area of nanomedicine development with great potential for AIT is the design of nanocarriers with a particular biodistribution profile that enables selective organ delivery. In the case of AIT, there are some organs that are considered “tolerogenic”, since the microenvironment surrounding particular types of immune cells in these organs facilitates the generation of regulatory T cells (Tregs) or other types of specific tolerance-generating mechanisms. Examples of these organs are the liver [101] and the spleen [102], and, as recently proposed, potentially also the lungs [103,104]. In the context of liver-targeted delivery for AIT, Liu et al. prepared targeted PLGA nanoparticles loaded with the allergen OVA. Once systemically administered, these nanoparticles accumulated in the liver, where they released their cargo within liver sinusoidal endothelial cells (LSEC) (Figure 4) [101]. LSEC act as antigen-presenting cells that can lead to the generation of Treg cells. In vivo, this nanoparticle-based strategy led to an increase in the production of TGF-β, IL-4, and IL-10, and it also reduced the allergic response towards OVA when used as a prophylactic strategy.

### 3.3. Use of Bio- and Nano-Materials as Co-Delivery Systems Containing the Allergen and Immunomodulatory Molecules

The third (and more technically complex) option for immunotherapy is the use of the materials as co-delivery systems to transport the allergen to certain cells in combination with other molecules, such as immunosuppressive drugs as rapamycin that generate a tolerogenic microenvironment [105]. The idea behind this co-delivery strategy is to bias the cellular response towards the generation of tolerogenic dendritic cells and Treg cells, potentially enhancing AIT efficacy. This co-delivery approach can be combined with all of the strategies mentioned in the previous section regarding different types of cargos or administration routes.

One example of this co-delivery strategy was developed by Shahgordi et al., who prepared PLGA nanoparticles loaded with curcumin, OVA, or both [106] for SLIT in a mouse model of allergic rhinitis. OVA was, therefore, the allergen used for the immunotherapy, while curcumin would act as an immunomodulatory drug. The formulation produced a decrease in total IgE levels and eosinophil cell count, and an increased IFN-γ to IL-4 ratio when compared to the standard SCIT with OVA. The optimal treatment was found to be the combination of free curcumin with encapsulated OVA or free OVA with encapsulated curcumin, instead of the nanoformulation where both agents were co-loaded. In another example, Hong et al. prepared methoxy poly(ethylene glycol)-poly(D,L-lactide) (mPEG-PDLLA) nanoparticles to co-deliver the peptide IK (an OVA epitope fragment) and the adjuvant R848, which is a Toll-like receptor (TLR)-7 ligand [107]. As in the previous case, the co-loaded molecule R848 was included in the formulation to modulate the immune response. Since oral administration was selected, the target cells that interacted with the nanoformulation to yield a therapeutic effect were intestinal DCs. The results obtained show the successful generation of tolerogenic intestinal DCs and the promotion of Tregs by the formulation, both in vitro and in vivo. This led to the inhibition of the allergic response in vivo, preventing the body temperature decrease and the appearance of diarrhea. This improvement in clinical symptoms was accompanied by a decrease in the levels of OVA-specific IgE, OVA-specific IgG1, IL-4, and IL-13, as well as an increase in the levels of OVA-specific IgG2a and IFN-γ [107].

A clear sample of the combination of co-delivery approaches can be found in a recent work by Yu et al. regarding MN arrays for peanut allergy immunotherapy (Figure 5) [108]. In this work, the authors prepared an MN formulation that dissolves once inserted in the skin, releasing a combination of three cargos within the superficial layers of the skin. The combined cargo consisted of peanut allergen, 1,25-dihydroxyvitamin D_3_ (VD3), and CpG oligonucleotide. The peanut allergen acts as the main agent responsible for the immunotherapy, while the other two components (VD3 and CpG) act as modulators of the immune response. On one side, VD3 is an immunosuppressant that has been shown to bias the immune responses towards a tolerogenic profile. On the other hand, CpG is an agonist of TLR-9 driving Th1 responses, and CpG oligodeoxynucleotide nanomedicines have been thoroughly studied for allergy treatments, among other therapeutic applications [109]. The combination of these two agents was selected to drive the immune response away from the Th2 profile. In a mouse model of peanut allergy, this co-delivery system led to a decrease in allergy scores, specific IgE, and intestinal and mucosal MC, and eosinophils. Furthermore, this was accompanied by an increase in the levels of IL-10, TGF-β, and Treg-like cells.

Regarding the comparison of tolerogenic co-delivery with other strategies, Liu et al. recently reported the efficacy of LSEC-targeted (liver-accumulating) OVA-loaded PLGA nanoparticles compared with non-targeted OVA-loaded PLGA nanoparticles that also include one immunomodulatory agent, either curcumin or rapamycin [110]. Similar efficacy of LSEC-targeted formulations and non-targeted RAPA/OVA co-loaded nanoparticles was found regarding OVA-specific antibodies (IgE, IgG1, and IgG2a) as well as IL-4, IL-5, IL-10, TGF-β, and IFN-γ levels. Furthermore, these nanoparticles showed significant therapeutic efficacy in two different mouse models sensitized with OVA: allergic airway disease and anaphylaxis.

To end this section of the article, Table 1 includes the main characteristics of the different therapeutic strategies described here for different allergic disease models using bio- or nano-materials.

## 4. Conclusions

The key question in novel allergy therapy approaches is how to maintain and restore tolerance. Specific immunotherapy has evolved in many ways, and the reestablishment of tolerance is partially possible. The driving mechanism behind such therapeutic strategies is the induction of allergen-specific regulatory subsets of T and B cells. However, safety concerns in patients with severe reactions and low efficacy for some allergens highlight the need for more research in this area. For therapeutic use, cell therapies and bio- and nano-material-based strategies offer good promise for future development. A potential combination of both strategies, employing biomaterials for exogenous cell delivery in tolerance-generating therapy in the context of allergy, remains largely unexplored, although some works have already been performed in other areas, such as autoimmune diseases and allogeneic transplantation [111]. On the other hand, and while there are no established means for primary allergy prevention, bacterial products have recently shown some promise that deserves more thorough research.

## Figures and Tables

**Figure 1 pharmaceutics-13-02149-f001:**
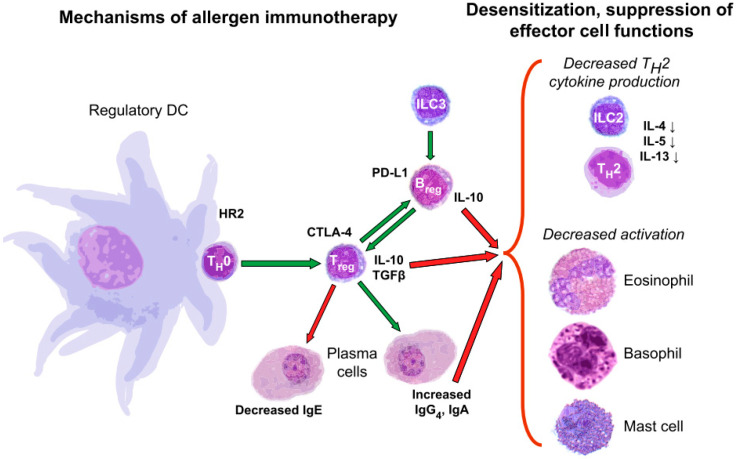
Schematic representation of the mechanism of tolerance generation by antigen-specific immunotherapy. This image is reproduced with permission from reference [5]. Copyright© 2021, Elsevier.

**Figure 2 pharmaceutics-13-02149-f002:**
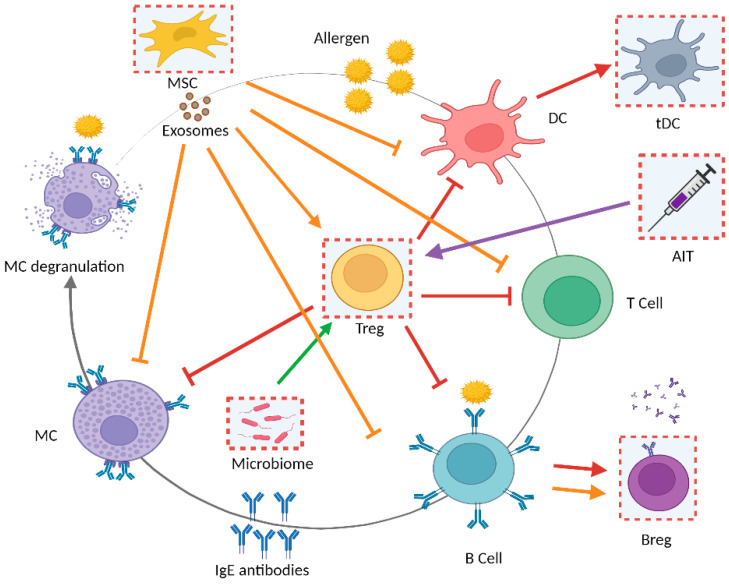
Schematic representation of the allergy players and the key points where the cell therapy and bio- or nano-material-based approaches described in this review could be used to achieve tolerance in allergy treatments. AIT, allergen-specific immunotherapy; B Cell, B lymphocytes; Breg, regulatory B lymphocytes; DCs, dendritic cells; IgE, allergen-specific immunoglobulin E; MCs, mast cells; MSC, mesenchymal stromal cells; tDCs, tolerogenic dendritic cells; Treg, regulatory T cells. Created with BioRender.com (7 December 2021).

**Figure 3 pharmaceutics-13-02149-f003:**
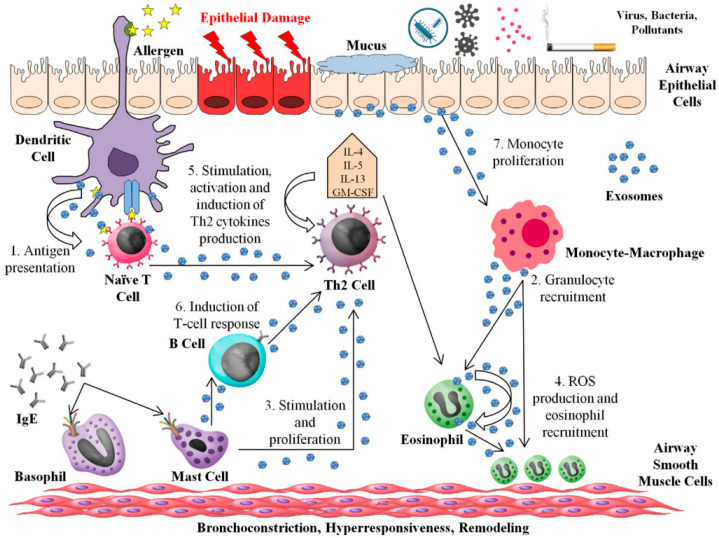
Schematic representation of the role of exosomes in allergy and asthma. This image is reproduced from reference [51]. The original publication was published under Creative Commons Attribution License (CC BY), 2017, Frontiers.

**Figure 4 pharmaceutics-13-02149-f004:**
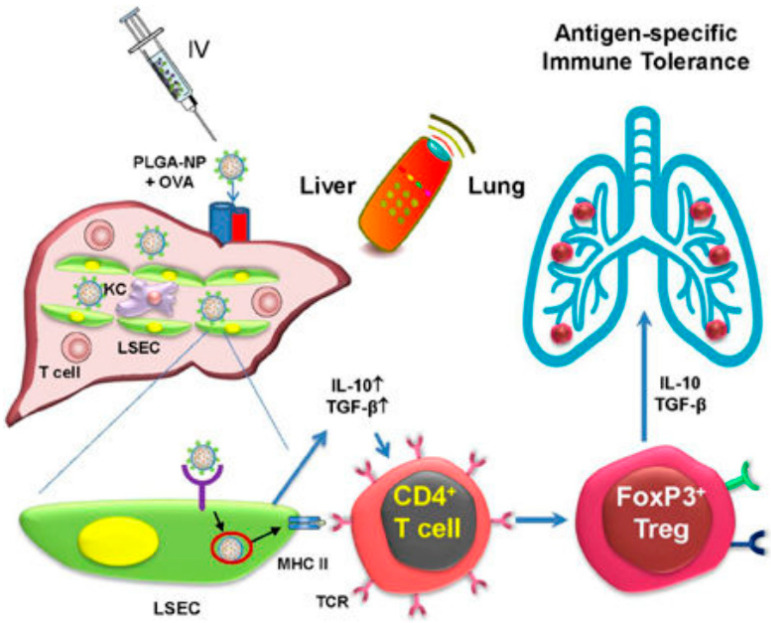
Schematic representation of the therapeutic strategy developed by Liu et al. for liver-targeted nanomedicine for AIT. This image is reproduced with permission from reference [101]. Copyright© 2021, American Chemical Society.

**Figure 5 pharmaceutics-13-02149-f005:**
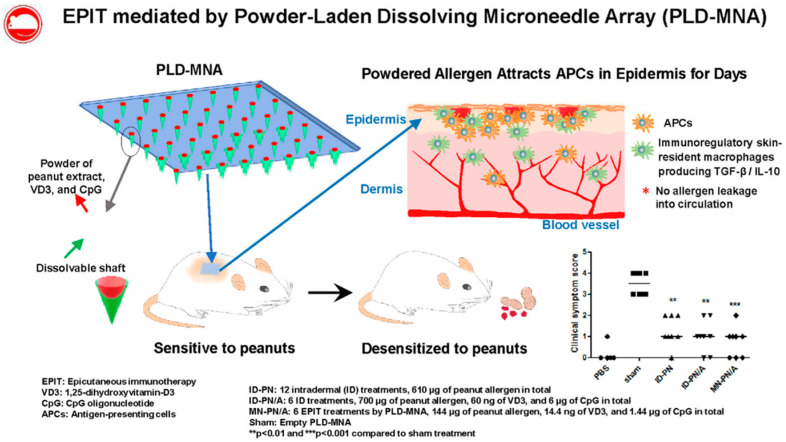
Schematic representation of the co-delivery strategy developed by Yu et al. for microneedle-mediated AIT. This image is reproduced with permission from reference [108]. Copyright© 2021, Elsevier.

**Table 1 pharmaceutics-13-02149-t001:** Main characteristics of the bio- or nano-material-based strategies for allergy therapy described in the article, in the order in which they appear in the text.

Bio- or Nano-Material	Experimental Model	Therapeutic Mechanism	Administration Route	Specific Cell Targeting	Cargo	Ref.
PVA-SPIONs	Incubation with CD4+ T cells in vitro	Direct decrease in antigen processing	N/A (in vitro incubation)	No	None	[83]
Fullerenes	Passive anaphylaxis mouse model (DNP as hapten allergen)	Direct effect on MC, decreasing IgE-induced release of mediators	Intraperitoneal injection	No	None	[84]
Polystyrene and PLGA nanoparticles	Allergic airway inflammation mouse model (OVA as allergen)	Immunotherapy through allergen delivery	Intravenous injection	No	OVA (conjugated or encapsulated)	[89]
PEG acetal dimethacrylate nanoparticles	Incubation with DCs co-cultured with T cells in vitro	Immunotherapy through allergen delivery (pH-cleavable carrier)	N/A (in vitro incubation)	No	OVA, grass pollen extract, dust mite allergen.	[93]
Poly(anhydride) nanoparticles	Particle administration to non-sensitized mice	Immunotherapy through allergen delivery	Intradermal injection	No	Peanut extract	[94]
Dendrosomes	Prophylactic use in mice, prior to sensitization with rBetv1	Indirect immunotherapy through delivery of plasmid encoding allergen	Footpad injection	No	Plasmid DNA encoding Betv 1a	[95]
PLGA nanoparticles	OVA-induced allergic rhinitis mouse model	Immunotherapy through allergen delivery	Sublingual	DC-targeted with aptamer	OVA	[96]
Dendrimer	Pru p 3-induced anaphylaxis mouse model	Immunotherapy through allergen delivery	Sublingual	DC-targeted with mannose	Pru p 3 peptide	[97]
Gas-filled microbubbles	OVA-induced allergic asthma mouse model	Immunotherapy through allergen delivery	Intranasal	No	OVA	[99]
Hyaluronate-based microneedle patches	Atopic dermatitis mouse model	Immunotherapy through allergen delivery	Epicutaneous	No	Der f1 dust-mite allergen	[100]
PLGA nanoparticles	OVA-induced allergic airway disease mouse model	Immunotherapy through allergen delivery	Intravenous injection	LSEC-targeted with mannan or peptide	OVA	[101]
PLGA nanoparticles	OVA-induced allergic rhinitis mouse model	Immunotherapy through co-delivery of allergen and modulatory molecules	Sublingual	No	Curcumin and OVA	[106]
mPEG-PDLLA nanoparticles	OVA-induced food allergy model	Immunotherapy through co-delivery of allergen and modulatory molecules	Oral	No	Peptide IK (OVA fragment) and R848 (TLR-7 ligand)	[107]
Dissolving microneedle patches	Peanut allergy mouse model	Immunotherapy through co-delivery of allergen and modulatory molecules	Epicutaneous	No	Peanut allergen, VD3, and CpG oligonucleotide	[108]
PLGA nanoparticles	OVA-induced allergic airway disease and OVA-induced anaphylaxis mouse models	Immunotherapy through co-delivery of allergen and modulatory molecules	Intravenous injection	No (comparison with LSEC-targeted without co-delivery)	OVA plus rapamycin or curcumin	[110]

## Data Availability

Not applicable.

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
