# Peer review of "New Therapeutic Approaches for Allergy: A Review of Cell Therapy and Bio- or Nano-Material-Based Strategies"

_pharmaceutics, 2021, doi:10.3390/pharmaceutics13122149_

Round 1

Reviewer 1 Report

All four drawings presented by the authors in the manuscript are reproductions of drawings already published by other authors. Authors should propose their drawing / diagram summarizing the studies they describe in the submitted manuscript.

The summary table would also facilitate the view of the collected results by organizing them

This review article described novel allergy therapy approaches.

The " Methods " part with flow diagram is missing. Flow diagram should contain information which were the keyword when searching articles and how many of them were found.

On page 2, in the sentence defining the purpose of the work, two approaches are assumed: cell therapy and bio or nano- material based therapies, however, later in the manuscript there are three part: 2,3 and 4. This should be unified

All four drawings presented by the authors in the manuscript are reproductions of drawings already published by other authors. Authors should propose their drawing / diagram summarizing the studies they describe in the submitted manuscript.

The summary table would also facilitate the view of the collected results by organizing them

Minor remarks:

Page 3, unnecessary use of an oblique font: “them to produce”

Page 8, line 3 and 8, two dots were placed instead of one

Author Response

Response to Reviewer 1:

Response: We thank the reviewer for their comments, aimed at improving the quality of our work.

Comment: All four drawings presented by the authors in the manuscript are reproductions of drawings already published by other authors. Authors should propose their drawing / diagram summarizing the studies they describe in the submitted manuscript.

According to the reviewer’s comment, we have included a new Figure (Figure 2) made by us, where we made a schematic representation of the allergy players and the key points where the cell therapy and bio- or nano-material based approaches described in this review could be used to achieve tolerance in allergy treatments.

Comment: The summary table would also facilitate the view of the collected results by organizing them

Response: According to the reviewer’s comment, we have included a new table (Table 1), summarizing all the articles discussed in section 3 of the manuscript.

This review article described novel allergy therapy approaches.

Comment: The " Methods " part with flow diagram is missing. Flow diagram should contain information which were the keyword when searching articles and how many of them were found.

Response: As stated in the revised version of our manuscript, “This review does not aim to provide a comprehensive list of all the novel strategies for allergy therapy that can be found in the literature. The main purpose of this work is to describe the main types of cell-, bio- or nano-material based strategies de-signed to acquire tolerance as allergy treatment (Figure 2). Relevant examples will be highlighted to illustrate each of the main strategies within each of these areas”. Given that this work is neither a meta-analysis nor a comprehensive review, we believe that a flow diagram as the one the reviewer is proposing would not fit with the actual methodology employed to produce this work.

Comment: On page 2, in the sentence defining the purpose of the work, two approaches are assumed: cell therapy and bio or nano- material based therapies, however, later in the manuscript there are three part: 2,3 and 4. This should be unified

Response: We thank the reviewer for pointing out this issue. According to the reviewer’s comment, both cell-focused sections have been focused within one larger section: “2. Cell-based therapeutic actions toward tolerance state in allergy”: “2.1 Key players involved in allergy and their use for cell therapy” and “2.2. Other cell-based therapeutic actions in allergy”.

Comment: All four drawings presented by the authors in the manuscript are reproductions of drawings already published by other authors. Authors should propose their drawing / diagram summarizing the studies they describe in the submitted manuscript.

Response: As stated in the first response, according to the reviewer’s comment, we have included a new Figure (Figure 2) made by us, where we provide a schematic representation of the manuscript content.

Comment: The summary table would also facilitate the view of the collected results by organizing them

Response: As stated in the second response, according to the reviewer’s comment, we have included a new table (Table 1), summarizing all the articles discussed in section 3 of the manuscript.

Minor remarks:

Page 3, unnecessary use of an oblique font: “them to produce”

Page 8, line 3 and 8, two dots were placed instead of one

Response: We thank the reviewer for pointing out these issues. They have been fixed in the revised manuscript.

Reviewer 2 Report

The manuscript has a unique point of view that would be most promising in allergy therapy. Nanoscale drug therapy is a well-studied area, but immune-modulating based allergy treatment can be an exciting area of research. Overall the review is well organized. There is little concern that needs to highlight in the revision submission before acceptance. It including:

  1. In the abstract 'severe reaction,' what kinds of reactions need clarification for the general reader? For example, it could be a severe allergic reaction.
  2. In the Introduction section, the nanomaterial-based therapies seem to miss which need to discuss.
  3. The concept and idea about 'nanomaterial-based strategies' have to describe clearly. Does the author mean nanomaterial-based carrier for allergy therapy, or is it nano-agents development for allergy therapy?
  4. The authors could add a list or table for different nanomaterial uses in allergy therapy, such as PLGA nanoparticles and micro needle-based systems.
  5. The authors could consider adding a list of abbreviations if possible in the manuscript scope of the publication. It would be helpful for the reader.

Author Response

Response to Reviewer 2:

Response: We thank the reviewer for their comments, aimed at improving the quality of our work.

Comment: The manuscript has a unique point of view that would be most promising in allergy therapy. Nanoscale drug therapy is a well-studied area, but immune-modulating based allergy treatment can be an exciting area of research. Overall the review is well organized. There is little concern that needs to highlight in the revision submission before acceptance. It including:

Comment: 1. In the abstract 'severe reaction,' what kinds of reactions need clarification for the general reader? For example, it could be a severe allergic reaction.

Response: We thank the reviewer for pointing out this issue. The text in the revised manuscript has been modified to state “severe allergic reaction”.

Comment: 2. In the Introduction section, the nanomaterial-based therapies seem to miss which need to discuss.

Response: According to the reviewer’s comment, we have included a new paragraph in the introduction regarding nanomaterial-based therapies.

“Particularly, the use of nanomaterials for biomedical application (nanomedicine) has attracted great interest in recent years [6]. The early development of the nanomedicine field focused mainly on the use of drug delivery nanocarriers for anti-cancer therapy [7]. This focus on cancer nanomedicine was mainly driven by the accumulation of nanoparticles in tumor tissue due to what was called the enhanced permeability and retention (EPR) effect [8,9]. The EPR effect was then used as the main rationale behind cancer nanomedicine and was also commonly referred to as the “passive targeting” principle. Further progress in the field lead to the development of “active” targeting approaches, where the surface of nanoparticles was decorated with different moieties driving enhanced uptake by certain target cell types [10]. Although the physiological mechanism behind passive accumulation of nanoparticles in tumors has been called into question [11], the tools that have been devised by the cancer nanomedicine community are now being exploited for other purposes. Among the most promising applications currently being evaluated is the use of nano-particles for immunotherapy in cancer [12] as well as in other diseases such as auto-immunity and organ transplantation [13–15]. Here we will focus in a promising (although still relatively underexplored) use of bio- and nano-material for immunomodulation: allergy therapy.”

Comment: 3. The concept and idea about 'nanomaterial-based strategies' have to describe clearly. Does the author mean nanomaterial-based carrier for allergy therapy, or is it nano-agents development for allergy therapy?

Response: According to the reviewer’s comment, we have included a new sentence in the Introduction clarifying this issue.

“These bio- and nano-material based strategies include approaches in which the material interacts directly with immune cells to provide a therapeutic effect, as well as schemes in which the material acts as a delivery system for an allergen, a drug or a combination of both”

Comment: 4. The authors could add a list or table for different nanomaterial uses in allergy therapy, such as PLGA nanoparticles and micro needle-based systems.

Response: According to the reviewer’s comment, we have included a new table (Table 1), summarizing all the articles discussed in section 3 of the manuscript.

Comment: 5. The authors could consider adding a list of abbreviations if possible in the manuscript scope of the publication. It would be helpful for the reader.

Response: According to the reviewer’s comment, we have included a list of abbreviations.

Reviewer 3 Report

In this paper, the authors reviewed the therapeutic approaches for allergy, which is interesting and of importance, the followings are some comments to this paper.

1.“Using bio and nanomaterials with a direct effect on cells involved in the allergic response by their inherent materials properties (line 372)”- Could you please put specific example of such materials and properties? Later on, you have mentioned about fullerene. What are the inherent properties of Carbon-based materials those can induce such effects?

2. “By targeting nanocarriers towards specific cell types, the needed dose of allergen decreased (line 417)”- Could you explain a bit more in terms of nanoparticles’ properties and mechanism such as “EPR effect”.

3. Some typos are noticeable (e.g. line no. 326, 331, 389, 439, 460, 496, 505, 536, 556, 607).

4. A table summarizing the nano-based therapeutic approaches focusing on biomaterials, allergen, advantage and rationale of nano formulations and some other properties could make this article more useful.

Author Response

Response to Reviewer 3:

Response: We thank the reviewer for their comments, aimed at improving the quality of our work.

Comment: In this paper, the authors reviewed the therapeutic approaches for allergy, which is interesting and of importance, the followings are some comments to this paper.

Comment: 1.“Using bio and nanomaterials with a direct effect on cells involved in the allergic response by their inherent materials properties (line 372)”- Could you please put specific example of such materials and properties? Later on, you have mentioned about fullerene. What are the inherent properties of Carbon-based materials those can induce such effects?

Response: According to the reviewer’s comment, we have modified the text to clarify this issue. While it is not clear which specific physicochemical characteristics of the material are responsible for the observed effect on immune cells, the main point of this section is to highlight approaches in which the material is directly responsible for the biological effect, in opposition to those in which the material acts as a delivery system for allergens and/or drugs. The new text states:

“these materials can also inherently act on the allergic response (either positively or negatively) without the need for a therapeutic cargo”

“In this therapeutic strategy, the effect is caused directly by the interaction of the material with a certain cell type involved in the allergic response, without any cargo being delivered from the material.”

Thus, the section is now called “Using bio- and nano-materials with a direct effect on cells involved in the allergic response”.

Comment: 2. “By targeting nanocarriers towards specific cell types, the needed dose of allergen decreased (line 417)”- Could you explain a bit more in terms of nanoparticles’ properties and mechanism such as “EPR effect”.

Response: According to the reviewer’s comment, we have included a new paragraph in the introduction regarding nanomaterial-based therapies.

“Particularly, the use of nanomaterials for biomedical application (nanomedicine) has attracted great interest in recent years [6]. The early development of the nanomedicine field focused mainly on the use of drug delivery nanocarriers for anti-cancer therapy [7]. This focus on cancer nanomedicine was mainly driven by the accumulation of nanoparticles in tumor tissue due to what was called the enhanced permeability and retention (EPR) effect [8,9]. The EPR effect was then used as the main rationale behind cancer nanomedicine and was also commonly referred to as the “passive targeting” principle. Further progress in the field lead to the development of “active” targeting approaches, where the surface of nanoparticles was decorated with different moieties driving enhanced uptake by certain target cell types [10]. Although the physiological mechanism behind passive accumulation of nanoparticles in tumors has been called into question [11], the tools that have been devised by the cancer nanomedicine community are now being exploited for other purposes. Among the most promising applications currently being evaluated is the use of nano-particles for immunotherapy in cancer [12] as well as in other diseases such as auto-immunity and organ transplantation [13–15]. Here we will focus in a promising (although still relatively underexplored) use of bio- and nano-material for immunomodulation: allergy therapy.”

Comment: 3. Some typos are noticeable (e.g. line no. 326, 331, 389, 439, 460, 496, 505, 536, 556, 607).

Response: We thank the reviewer for pointing out these issues. They have been fixed in the revised manuscript.

  1. A table summarizing the nano-based therapeutic approaches focusing on biomaterials, allergen, advantage and rationale of nano formulations and some other properties could make this article more useful.

Response: According to the reviewer’s comment, we have included a new table (Table 1), summarizing all the articles discussed in section 3 of the manuscript.

Round 2

Reviewer 1 Report

I have no more comments

Reviewer 2 Report

The authors have done all the required corrections on their revised manuscript. The editor could consider accepting the review manuscript.